# The Robustness of Estimator Composition

**Pingfan Tang**
School of Computing
University of Utah
Salt Lake City, UT 84112
tang1984@cs.utah.edu

**Jeff M. Phillips**
School of Computing
University of Utah
Salt Lake City, UT 84112
jeffp@cs.utah.edu

## Abstract

We formalize notions of robustness for composite estimators via the notion of a breakdown point. A composite estimator successively applies two (or more) estimators: on data decomposed into disjoint parts, it applies the first estimator on each part, then the second estimator on the outputs of the first estimator. And so on, if the composition is of more than two estimators. Informally, the breakdown point is the minimum fraction of data points which if significantly modified will also significantly modify the output of the estimator, so it is typically desirable to have a large breakdown point. Our main result shows that, under mild conditions on the individual estimators, the breakdown point of the composite estimator is the product of the breakdown points of the individual estimators. We also demonstrate several scenarios, ranging from regression to statistical testing, where this analysis is easy to apply, useful in understanding worst case robustness, and sheds powerful insights onto the associated data analysis.

## 1 Introduction

Robust statistical estimators [5, 7] (in particular, resistant estimators), such as the median, are an essential tool in data analysis since they are provably immune to outliers. Given data with a large fraction of extreme outliers, a robust estimator guarantees the returned value is still within the non-outlier part of the data. In particular, the role of these estimators is quickly growing in importance as the scale and automation associated with data collection and data processing becomes more commonplace. Artisanal data (hand crafted and carefully curated), where potential outliers can be removed, is becoming proportionally less common. Instead, important decisions are being made blindly based on the output of analysis functions, often without looking at individual data points and their effect on the outcome. Thus using estimators as part of this pipeline that are not robust are susceptible to erroneous and dangerous decisions as the result of a few extreme and rogue data points.

Although other approaches like regularization and pruning a constant number of obvious outliers are common as well, they do not come with the important guarantees that ensure these unwanted outcomes absolutely cannot occur.

In this paper we initiate the formal study of the robustness of composition of estimators through the notion of breakdown points. These are especially important with the growth of data analysis pipelines where the final result or prediction is the result of several layers of data processing. When each layer in this pipeline is modeled as an estimator, then our analysis provides the first general robustness analysis of these processes.

The *breakdown point* [4, 3] is a basic measure of robustness of an estimator. Intuitively, it describes how many outliers can be in the data without the estimator becoming unreliable. However, the literature is full of slightly inconsistent and informal definitions of this concept. For example:

- Aloupis [1] write "the breakdown point is the proportion of data which must be moved to infinity so that the estimator will do the same."

- Huber and Ronchetti [8] write "the breakdown point is the smallest fraction of bad observations that may cause an estimator to take on arbitrarily large aberrant values."

- Dasgupta, Kumar, and Srikumar [14] write "the breakdown point of an estimator is the largest fraction of the data that can be moved arbitrarily without perturbing the estimator to the boundary of the parameter space."

All of these definitions have similar meanings, and they are typically sufficient for the purpose of understanding a single estimator. However, they are not mathematically rigorous, and it is difficult to use them to discuss the breakdown point of composite estimators.

**Composition of Estimators.** In a bit more detail (we give formal definitions in Section 2.1), an estimator $E$ maps a data set to single value in another space, sometimes the same as a single data point. For instance the mean or the median are simple estimators on one-dimensional data. A composite $E_1$-$E_2$ estimator applies two estimators $E_1$ and $E_2$ on data stored in a hierarchy. Let $\mathcal{P} = \{P_1, P_2, \ldots, P_n\}$ be a set of subdata sets, where each subdata set $P_i = \{p_{i,1}, p_{i,2}, \ldots, p_{i,k}\}$ has individual data readings. Then the $E_1$-$E_2$ estimator reports $E_2(E_1(P_1), E_1(P_2), \ldots, E_1(P_n))$, that is the estimator $E_2$ applied to the output of estimator $E_1$ on each subdata set.

## 1.1 Examples of Estimator Composition

Composite estimators arise in many scenarios in data analysis.

**Uncertain Data.** For instance, in the last decade there has been increased focus on the study of uncertainty data [10, 9, 2] where instead of analyzing a data set, we are given a model of the uncertainty of each data point. Consider tracking the summarization of a group of $n$ people based on noisy GPS measurements. For each person $i$ we might get $k$ readings of their location $P_i$, and use these $k$ readings as a discrete probability distribution of where that person might be. Then in order to represent the center of this set of people a natural thing to do would be to estimate the location of each person as $x_i \leftarrow E_1(P_i)$, and then use these estimates to summarize the entire group $E_2(x_1, x_2, \ldots, x_n)$. Using the mean as $E_1$ and $E_2$ would be easy, but would be susceptible to even a single outrageous outlier (all people are in Manhattan, but a spurious reading was at $(0,0)$ lat-long, off the coast of Africa). An alternative is to use the $L_1$-median for $E_1$ and $E_2$, that is known to have an optimal breakdown point of $0.5$. But what is the breakdown point of the $E_1$-$E_2$ estimator?

**Robust Analysis of Bursty Behavior.** Understanding the robustness of estimators can also be critical towards how much one can "game" a system. For instance, consider a start-up media website that gets bursts of traffic from memes they curate. They publish a statistic showing the median of the top half of traffic days each month, and aggregate these by taking the median of such values over the top half of all months. This is a composite estimator, and they proudly claim, even through they have bursty traffic, it is robust (each estimator has a breakdown point of $0.25$). If this composite estimator shows large traffic, should a potential buyer of this website by impressed? Is there a better, more robust estimator the potential buyer could request? If the media website can stagger the release of its content, how should they distribute it to maximize this composite estimator?

**Part of the Data Analysis Pipeline.** This process of estimator composition is very common in broad data analysis literature. This arises from the idea of an "analysis pipeline" where at several stages estimators or analysis is performed on data, and then further estimators and analysis are performed downstream. In many cases a robust estimator like the median is used, specifically for its robustness properties, but there is no analysis of how robust the composition of these estimators is.

## 1.2 Main Results

This paper initiates the formal and general study of the robustness of composite estimators.

- In Subsection 2.1, we give two formal definitions of breakdown points which are both required to prove composition theorem. One variant of the definition closely aligns with other formalizations [4, 3], while another is fundamentally different.

- The main result provides general conditions under which an $E_1$-$E_2$ estimator with breakdown points $\beta_1$ and $\beta_2$, has a breakdown point of $\beta_1\beta_2$ (Theorem 2 in Subsection 2.2).

- Moreover, by showing examples where our conditions do not strictly apply, we gain an understanding of how to circumvent the above result. An example is in composite percentile estimators (e.g., $E_1$ returns the 25th percentile, and $E_2$ the 75th percentile of a ranked set). These composite estimators have larger breakdown point than $\beta_1 \cdot \beta_2$.

- The main result can extended to multiple compositions, under suitable conditions, so for instance an $E_1$-$E_2$-$E_3$ estimator has a breakdown point of $\beta_1 \beta_2 \beta_3$ (Theorem 3 in Subsection 2.3). This implies that long analysis chains can be very suspect to a few carefully places outliers since the breakdown point decays exponentially in the length of the analysis chain.

- In Section 3, we highlight several applications of this theory, including robust regression, robustness of p-values, a depth-3 composition, and how to advantageously manipulate the observation about percentile estimator composition. We demonstrate a few more applications with simulations in Section 4.

## 2 Robustness of Estimator Composition

### 2.1 Formal Definitions of Breakdown Points

In this paper, we give two definitions for the breakdown point: *Asymptotic Breakdown Point* and *Asymptotic Onto-Breakdown Point*. The first definition, Asymptotic Breakdown Point, is similar to the classic formal definitions in [4] and [3] (including their highly technical nature), although their definitions of the estimator are slightly different leading to some minor differences in special cases. However our second definition, Asymptotic Onto-Breakdown Point, is a structurally new definition, and we illustrate how it can result in significantly different values on some common and useful estimators. Our main theorem will require both definitions, and the differences in performance will lead to several new applications and insights.

We define an *estimator $E$* as a function from the collection of some finite subsets of a metric space $(\mathscr{X}, d)$ to another metric space $(\mathscr{X}', d')$:

$$E : \mathscr{A} \subset \{X \subset \mathscr{X} \mid 0 < |X| < \infty\} \mapsto \mathscr{X}', \tag{1}$$

where $X$ is a multiset. This means if $x \in X$ then $x$ can appear more than once in $X$, and the multiplicity of elements will be considered when we compute $|X|$.

**Finite Sample Breakdown Point.** For estimator $E$ defined in (1) and positive integer $n$ we define its *finite sample breakdown point $g_E(n)$* over a set $M$ as

$$g_E(n) = \begin{cases} \max(M) & \text{if } M \neq \emptyset \\ 0 & \text{if } M = \emptyset \end{cases} \tag{2}$$

where for $\rho(x', X) = \max_{x \in X} d(x', x)$ is the distance from $x'$ to the furthest point in $X$,

$$M = \{m \in [0, n] \mid \forall X \in \mathscr{A}, |X| = n, \forall\, G_1 > 0, \exists\, G_2 = G_2(X, G_1) \text{ s.t. } \forall X' \in \mathscr{A},$$
$$\text{if } |X'| = n \text{ and } |\{x' \in X' \mid \rho(x', X) > G_1\}| \leq m \text{ then } d'(E(X), E(X')) \leq G_2\}. \tag{3}$$

For an estimator $E$ in (1) and $X \in \mathscr{A}$, the finite sample breakdown point $g_E(n)$ means if the number of unbounded points in $X'$ is at most $g_E(n)$, then $E(X')$ will be bounded. Lets break this definition down a bit more. The definition holds over all data sets $X \in \mathscr{A}$ of size $n$, and for all values $G_1 > 0$ and some value $G_2$ defined as a function $G_2(X, G_1)$ of the data set $X$ and value $G_1$. Then $g_E(n)$ is the maximum value $m$ (over all $X$, $G_1$, and $G_2$ above) such that for all $X' \in \mathscr{A}$ with $|X'| = n$ then $|\{x' \in X' \mid \rho(x', X) > G_1\}| \leq m$ (that is at most $m$ points are further than $G_1$ from $X$) where the estimators are close, $d'(E(X), E(X')) \leq G_2$.

For example, consider a point set $X = \{0, 0.15, 0.2, 0.25, 0.4, 0.55, 0.6, 0.65, 0.72, 0.8, 1.0\}$ with $n = 11$ and median 0.55. If we set $G_1 = 3$, then we can consider sets $X'$ of size 11 with fewer than $m$ points that are either greater than 3 or less than $-2$. This means in $X'$ there are at most $m$ points which are greater than 3 or less than $-2$, and all other $n - m$ points are in $[-2, 3]$. Under these conditions, we can (conservatively) set $G_2 = 4$, and know that for values of $m$ as 1, 2, 3, 4, or 5, then the median of $X'$ must be between $-3.45$ and $4.55$; and this holds no matter where we set those $m$ points (e.g., at 20 or at 1000). This does not hold for $m \geq 6$, so $g_E(11) = 5$.

**Asymptotic Breakdown Point.** If the limit $\lim_{n\to\infty} \frac{g_E(n)}{n}$ exists, then we define this limit

$$\beta = \lim_{n\to\infty} \frac{g_E(n)}{n} \tag{4}$$

as the *asymptotic breakdown point*, or *breakdown point* for short, of the estimator $E$.

*Remark* 1. It is not hard to see that many common estimators satisfy the conditions. For example, the median, $L_1$-median [1], and Siegel estimators [11] all have asymptotic breakdown points of $0.5$.

**Asymptotic Onto-Breakdown Point.** For an estimator $E$ given in (1) and positive integer $n$, if $\widetilde{M} = \{0 \le m \le n \mid \forall\, X \in \mathscr{A}, |X| = n, \forall\, y \in \mathscr{X}', \exists\, X' \in \mathscr{A} \text{ s.t. } |X'| = n, |X \cap X'| = n - m, E(X') = y\}$ is not empty, we define

$$f_E(n) = \min(\widetilde{M}). \tag{5}$$

The definition of $f_E(n)$ implies, if we change $f_E(n)$ elements in $X$, we can make $E$ become *any* value in $\mathscr{X}'$: it is onto. In contrast $g_E(n)$ only requires $E(X')$ to become far from $E(X)$, perhaps only in one direction. Then the *asymptotic onto-breakdown point* is defined as the following limit if it exists

$$\lim_{n\to\infty} \frac{f_E(n)}{n}. \tag{6}$$

*Remark* 2. For a quantile estimator $E$ that returns a percentile other than the 50th, then $\lim_{n\to\infty} \frac{g_E(n)}{n} \ne \lim_{n\to\infty} \frac{f_E(n)}{n}$. For instance, if $E$ returns the 25th percentile of a ranked set, setting only 25% of the data points to $-\infty$ causes $E$ to return $-\infty$; hence $\lim_{n\to\infty} \frac{g_E(n)}{n} = 0.25$. And while any value less than the original 25th percentile can also be obtained; to return a value larger than the largest element in the original set, at least 75% of the data must be modified, thus $\lim_{n\to\infty} \frac{f_E(n)}{n} = 0.75$.

As we will observe in Section 3, this nuance in definition regarding percentile estimators will allow for some interesting composite estimator design.

## 2.2 Definition of $E1$-$E2$ Estimators, and their Robustness

We consider the following two estimators:

$$E_1: \mathscr{A}_1 \subset \{X \subset \mathscr{X}_1 \mid 0 < |X| < \infty\} \mapsto \mathscr{X}_2, \tag{7}$$
$$E_2: \mathscr{A}_2 \subset \{X \subset \mathscr{X}_2 \mid 0 < |X| < \infty\} \mapsto \mathscr{X}_2', \tag{8}$$

where any finite subset of $E_1(\mathscr{A}_1)$, the range of $E_1$, belongs to $\mathscr{A}_2$. Suppose $P_i \in \mathscr{A}_1, |P_i| = k$ for $i = 1, 2, \cdots, n$ and $P_{\text{flat}} = \uplus_{i=1}^n P_i$, where $\uplus$ means if $x$ appears $n_1$ times in $X_1$ and $n_2$ times in $X_2$ then $x$ appears $n_1 + n_2$ times in $X_1 \uplus X_2$. We define

$$E(P_{\text{flat}}) = E_2\left(E_1(P_1), E_1(P_2), \cdots, E_1(P_n)\right). \tag{9}$$

**Theorem 1.** *Suppose $g_{E_1}(k)$ and $g_{E_2}(n)$ are the finite sample breakdown points of estimators $E_1$ and $E_2$ which are given by (7) and (8) respectively. If $g_E(nk)$ is the finite sample breakdown point of $E$ given by (9), then we have $g_{E_2}(n)g_{E_1}(k) \le g_E(nk)$. If $\beta_1 = \lim_{k\to\infty} \frac{g_{E_1}(k)}{k}$, $\beta_2 = \lim_{n\to\infty} \frac{g_{E_2}(n)}{n}$ and $\beta = \lim_{n,k\to\infty} \frac{g_E(nk)}{nk}$ all exist, then we have $\beta_1\beta_2 \le \beta$.*

The proof of Theorem 1 and other theorems can be found in the full version of this paper [12].

*Remark* 3. Under the condition of Theorem 1, we cannot guarantee $\beta = \beta_1\beta_2$. For example, suppose $E_1$ and $E_2$ take the 25th percentile and the 75th percentile of a ranked set of real numbers respectively. So, we have $\beta_1 = \beta_2 = \frac{1}{4}$. However, $\beta = \frac{1}{4} \cdot \frac{3}{4} = \frac{3}{16}$.

In fact, the limit of $\frac{g_E(nk)}{nk}$ as $n, k \to \infty$ may even not exist. For example, suppose $E_1$ takes the 25th percentile of a ranked set of real numbers. When $n$ is odd $E_2$ takes the the 25th percentile of a ranked set of $n$ real numbers, and when $n$ is even $E_2$ takes the the 75th percentile of a ranked set of $n$ real numbers. Thus, $\beta_1 = \beta_2 = \frac{1}{4}$, but $g_E(nk) \approx \frac{1}{4}nk$ if $n$ is odd, and $g_E(nk) \approx \frac{1}{4} \cdot \frac{3}{4}nk$ if $n$ is even, which implies $\lim_{n,k\to\infty} \frac{g_E(nk)}{nk}$ does not exist.

Therefore, to guarantee $\beta$ exist and $\beta = \beta_1\beta_2$, we introduce the definition of asymptotic onto-breakdown point in (6). As shown in *Remark 2*, the values of (4) and (6) may be not equal. However, with the condition of the asymptotic breakdown point and asymptotic onto-breakdown point of $E_1$ being the same, we can finally state our desired clean result.

**Theorem 2.** *For estimators $E_1$, $E_2$ and $E$ given by (7), (8) and (9) respectively, suppose $g_{E_1}(k)$, $g_{E_2}(n)$ and $g_E(nk)$ are defined by (2), and $f_{E_1}(k)$ is defined by (5). Moreover, $E_1$ is an onto function and for any fixed positive integer $n$ we have*

$$\exists\, X \in \mathscr{A}_2, |X| = n, G_1 > 0, s.t. \; \forall\, G_2 > 0, \exists\, X' \in \mathscr{A}_2 \text{ satisfying}$$
$$|X'| = n, |X' \setminus X| = g_{E_2}(n) + 1, \text{ and } d_2'(E_2(X), E_2(X')) > G_2, \tag{10}$$

*where $d_2'$ is the metric of space $\mathscr{X}_2'$. If $\beta_1 = \lim_{k \to \infty} \frac{g_{E_1}(k)}{k} = \lim_{k \to \infty} \frac{f_{E_1}(k)}{k}$, and $\beta_2 = \lim_{n \to \infty} \frac{g_{E_2}(n)}{n}$ both exist, then $\beta = \lim_{n,k \to \infty} \frac{g_E(nk)}{nk}$ exists, and $\beta = \beta_1 \beta_2$.*

*Remark* 4. Without the introduction of $f_E(n)$, we cannot even guarantee $\beta \leq \beta_1$ or $\beta \leq \beta_2$ only under the condition of Theorem 1, even if $E_1$ and $E_2$ are both onto functions. For example, for any $P = \{p_1, p_2, \cdots, p_k\} \subset \mathbb{R}$ and $X = \{x_1, x_2, \cdots, x_n\} \subset \mathbb{R}$, we define $E_1(P) = 1/\text{median}(P)$ (if $\text{median}(P) \neq 0$, otherwise define $E_1(P) = 0$) and $E_2(X) = \text{median}(y_1, y_2, \cdots, y_n)$, where $y_i$ $(1 \leq y \leq n)$ is given by $y_i = 1/x_i$ (if $x_i \neq 0$, otherwise define $y_i = 0$). Since $g_{E_1}(k) = g_{E_2}(n) = 0$ for all $n, k$, we have $\beta_1 = \beta_2 = 0$. However, in order to make $E_2(E_1(P_1), E_1(P_2), \cdots, E_1(P_n)) \to +\infty$, we need to make about $\frac{n}{2}$ elements in $\{E(P_1), E(P_2), \cdots, E(P_n)\}$ go to $0+$. To make $E_1(P_i) \to 0+$, we need to make about $\frac{k}{2}$ points in $P_i$ go to $+\infty$. Therefore, we have $g_E(nk) \approx \frac{n}{2} \cdot \frac{k}{2}$ and $\beta = \frac{1}{4}$.

### 2.3 Multi-level Composition of Estimators

To study the breakdown point of composite estimators with more than two levels, we introduce the following estimator:

$$E_3 : \mathscr{A}_3 \subset \{X \subset \mathscr{X}_2' \mid 0 < |X| < \infty\} \mapsto \mathscr{X}_3', \tag{11}$$

where any finite subset of $E_2(\mathscr{A}_2)$, the range of $E_2$, belongs to $\mathscr{A}_3$. Suppose $P_{i,j} \in \mathscr{A}_1$, $|P_{i,j}| = k$ for $i = 1, 2, \cdots, n$, $j = 1, 2, \cdots, m$ and $P_{\text{flat}}^j = \biguplus_{i=1}^n P_{i,j}$, $P_{\text{flat}} = \biguplus_{j=1}^m P_{\text{flat}}^j$. We define

$$E(P_{\text{flat}}) = E_3 \left( E_2(\widetilde{P}_{\text{flat}}^1), E_2(\widetilde{P}_{\text{flat}}^2), \cdots, E_2(\widetilde{P}_{\text{flat}}^m) \right), \tag{12}$$

where $\widetilde{P}_{\text{flat}}^j = \{E_1(P_{1,j}), E_1(P_{2,j}), \cdots, E_1(P_{n,j})\}$, for $j = 1, 2, \cdots, m$.

From Theorem 2, we can obtain the following theorem about the breakdown point of $E$ in (12).

**Theorem 3.** *For estimators $E_1$, $E_2$, $E_3$ and $E$ given by (7), (8), (11) and (12) respectively, suppose $g_{E_1}(k)$, $g_{E_2}(n)$, $g_{E_3}(m)$ and $g_E(mnk)$ are defined by (2), and $f_{E_1}(k)$, $f_{E_2}(n)$ are defined by (5). Moreover, $E_1$ and $E_2$ are both onto functions, and for any fixed positive integer $m$ we have*

$$\exists\, X \in \mathscr{A}_3, |X| = m, G_1 > 0, s.t. \; \forall\, G_2 > 0, \exists\, X' \in \mathscr{A}_3$$
$$\text{satisfying } |X'| = m, |X' \setminus X| = g_{E_3}(m) + 1, \text{ and } d_3'(E_3(X), E_3(X')) > G_2,$$

*where $d_3'$ is the metric of space $\mathscr{X}_3'$. If $\beta_1 = \lim_{k \to \infty} \frac{g_{E_1}(k)}{k} = \lim_{k \to \infty} \frac{f_{E_1}(k)}{k}, \beta_2 = \lim_{n \to \infty} \frac{g_{E_2}(n)}{n} = \lim_{n \to \infty} \frac{f_{E_2}(n)}{n}$ and $\beta_3 = \lim_{m \to \infty} \frac{g_{E_3}(m)}{m}$ all exist, then we have $\beta = \lim_{m,n,k \to \infty} \frac{g_E(mnk)}{mnk}$ exists, and $\beta = \beta_1 \beta_2 \beta_3$ .*

## 3 Applications

### 3.1 Application 1 : Balancing Percentiles

For $n$ companies, for simplicity, assume each company has $k$ employees. We are interested in the income of the regular employees of all companies, not the executives who may have much higher pay. Let $p_{i,j}$ represents the income of the $j$th employee in the $i$th company. Set $P_{\text{flat}} = \biguplus_{i=1}^n P_i$ where the $i$th company has a set $P_i = \{p_{i,1}, p_{i,2}, \cdots, p_{i,k}\} \subset \mathbb{R}$ and for notational convenience $p_{i,1} \leq p_{i,2} \leq \cdots \leq p_{i,k}$ for $i \in \{1, 2, \cdots, n\}$. Suppose the income data $P_i$ of each company is preprocessed by a 45-percentile estimator $E_1$ (median of lowest 90% of incomes), with breakdown point $\beta_1 = 0.45$. In theory $E_1(P_i)$ can better reflect the income of regular employees in a company, since there may be about 10% of employees in the management of a company and their incomes are usually much higher than that of common employees. So, the preprocessed data is $X = \{E_1(P_1), E_1(P_2), \cdots, E_1(P_n)\}$.

If we define $E_2(X) = \text{median}(X)$ and $E(P_{\text{flat}}) = E_2(X)$, then the breakdown point of $E_2$ is $\beta_2 = 0.5$, and the breakdown points of $E$ is $\beta = \beta_1\beta_2 = 0.225$.

However, if we use another $E_2$, then $E$ can be more robust. For example, for $X = \{x_1, x_2, \cdots, x_n\}$ where $x_1 \leq x_2 \leq \cdots \leq x_n$, we can define $E_2$ as the 55-percentile estimator (median of largest 90% of incomes). In order to make $E(P_{\text{flat}}) = E_2(X) = E_2(E_1(P_1), E_1(P_2), \cdots, E_1(P_n))$ go to infinity, we need to either move 55% points of $X$ to $-\infty$ or move 45% points of $X$ to $+\infty$. In either case, we need to move about $0.45 \cdot 0.55nk$ points of $P_{\text{flat}}$ to infinity. This means the breakdown point of $E$ is $\beta = 0.45 \cdot 0.55 = 0.2475$ which is greater than $0.225$.

This example implies if we know how the raw data is preprocessed by estimator $E_1$, we can choose a proper estimator $E_2$ to make the $E_1$-$E_2$ estimator more robust.

## 3.2 Application 2 : Regression of $L_1$ Medians

Suppose we want to use linear regression to robustly predict the weight of a person from his or her height, and we have multiple readings of each person's height and weight. The raw data is $P_{\text{flat}} = \uplus_{i=1}^{n} P_i$ where for the $i$th person we have a set $P_i = \{p_{i,1}, p_{i,2}, \cdots, p_{i,k}\} \subset \mathbb{R}^2$ and $p_{i,j} = (x_{i,j}, y_{i,j})$ for $i \in \{1, 2, \cdots, n\}, j \in \{1, 2, \cdots, k\}$. Here, $x_{i,j}$ and $y_{i,j}$ are the height and weight respectively of the $i$th person in their $j$th measurement.

One "robust" way to process this data, is to first pre-process each $P_i$ with its $L_1$-median [1]: $(\bar{x}_i, \bar{y}_i) \leftarrow E_1(P_i)$, where $E_1(P_i) = L_1\text{-median}(P_i)$ has breakdown point $\beta_1 = 0.5$. Then we could generate a linear model to predict weight $\hat{y}_i = ax + b$ from the Siegel Estimator [11]: $E_2(Z) = (a, b)$, with breakdown point $\beta_2 = 0.5$. From Theorem 2 we immediately know the breakdown point of $E(P_{\text{flat}}) = E_2(E_1(P_1), E_1(P_2), \cdots, E_1(P_n))$ is $\beta = \beta_1\beta_2 = 0.5 \cdot 0.5 = 0.25$.

Alternatively, taking the Siegel estimator of $P_{\text{flat}}$ (i.e., returning $E_2(P_{\text{flat}})$) would have a much larger breakdown point of $0.5$. So a seemingly harmless operation of normalizing the data with a robust estimator (with optimal $0.5$ breakdown point) drastically decreases the robustness of the process.

## 3.3 Application 3 : Significance Thresholds

Suppose we are studying the distribution of the wingspread of fruit flies. There are $n = 500$ flies, and the variance of the true wingspread among these flies is on the order of $0.1$ units. Our goal is to estimate the $0.05$ significance level of this distribution of wingspread among normal flies.

To obtain a measured value of the wingspread of the $i$th fly, denoted $F_i$, we measure the wingspread of $i$th fly $k = 100$ times independently, and obtain the measurement set $P_i = \{p_{i,1}, p_{i,2}, \cdots, p_{i,k}\}$. The measurement is carried out by a machine automatically and quickly, which implies the variance of each $P_i$ is typically very small, perhaps only $0.0001$ units, but there are outliers in $P_i$ with small chance due to possible machine malfunction. This malfunction may be correlated to individual flies because of anatomical issues, or it may have autocorrelation (the machine jams for a series of consecutive measurements).

To perform hypothesis testing we desire the $0.05$ significance level, so we are interested in the 95th percentile of the set $F = \{F_1, F_2, \cdots, F_n\}$. So a post processing estimator $E_2$ returns the 95th percentile of $F$ and has a breakdown point of $\beta_2 = 0.05$ [6]. Now, we need to design an estimator $E_1$ to process the raw data $P_{\text{flat}} = \uplus_{i=1}^{n} P_i$ to obtain $F = \{F_1, F_2, \cdots, F_n\}$. For example, we can define $E_1$ as $F_i = E_1(P_i) = \text{median}(P_i)$ and estimator $E$ as $E(P_{\text{flat}}) = E_2(E_1(P_1), E_1(P_2), \cdots, E_1(P_n))$.

Then, the breakdown point of $E_1$ is $0.5$. Since the breakdown point of $E_2$ is $0.05$, the breakdown point of the composite estimator $E$ is $\beta = \beta_1\beta_2 = 0.5 \cdot 0.05 = 0.025$. This means if the measurement machine malfunctioned only $2.5\%$ of the time, we could have an anomalous significant level, leading to false discovery. Can we make this process more robust by adjusting $E_1$?

Actually, *yes!*, we can use another pre-processing estimator to get a more robust $E$. Since the variance of each $P_i$ is only $0.0001$, we can let $E_1$ return the 5th percentile of a ranked set of real numbers, then there is not much difference between $E_1(P_i)$ and the median of $P_i$. (Note: this introduces a small amount of bias that can likely be accounted for in other ways.) In order to make $E(P_{\text{flat}}) = E_2(F)$ go to infinity we need to move $5\%$ points of $X$ to $-\infty$ (causing $E_2$ to give an anomalous value) or $95\%$ points of $X$ to $+\infty$ (causing many, $95\%$, of the $E_1$ values, to give anomalous values). In either case, we need to move about $5\% \cdot 95\%$ points of $P_{\text{flat}}$ to infinity. So, the breakdown points of $E$ is

$\beta = 0.05 \cdot 0.95 = 0.0475$ which is greater than $0.025$. That is, we can now sustain up to $4.75\%$ of the measurement machine's reading to be anomalous, almost double than before, without leading to an anomalous significance threshold value.

This example implies if we know the post-processing estimator $E_2$, we can choose a proper method to preprocess the raw data to make the $E_1$-$E_2$ estimator more robust.

### 3.4 Application 4 : 3-Level Composition

Suppose we want to use a single value to represent the temperature of the US in a certain day. There are $m = 50$ states in the country. Suppose each state has $n = 100$ meteorological stations, and the station $i$ in state $j$ measures the local temperature $k = 24$ times to get the data $P_{i,j} = \{t_{i,j,1}, t_{i,j,2}, \cdots, t_{i,j,k}\}$. We define $P_{\text{flat}}^j = \uplus_{i=1}^n P_{i,j}$, $P_{\text{flat}} = \uplus_{j=1}^m P_{\text{flat}}^j$ and

$$E_1(P_{i,j}) = \text{median}(P_{i,j}), \quad E_2(P_{\text{flat}}^j) = \text{median}\left(E_1(P_{1,j}), E_1(P_{1,j}), \cdots, E_1(P_{n,j})\right)$$

$$E(P_{\text{flat}}) = E_3(E_2(P_{\text{flat}}^1), E_2(P_{\text{flat}}^2), \cdots, E_2(P_{\text{flat}}^m)) = \text{median}(E_2(P_{\text{flat}}^1), E_2(P_{\text{flat}}^2), \cdots, E_2(P_{\text{flat}}^m)).$$

So, the break down points of $E_1$, $E_2$ and $E_3$ are $\beta_1 = \beta_2 = \beta_3 = 0.5$. From Theorem 3, we know the break down point of $E$ is $\beta = \beta_1 \beta_2 \beta_3 = 0.125$. Therefore, we know the estimator $E$ is not very robust, and it may be not a good choice to use $E(P_{\text{flat}})$ to represent the temperature of the US in a certain day.

This example illustrates how the more times the raw data is aggregated, the more unreliable the final result can become.

## 4 Simulation: Estimator Manipulation

In this simulation we actually construct a method to relocate an estimator by modifying the smallest number of points possible. We specifically target the $L_1$-median of $L_1$-medians since its somewhat non-trivial to solve for the new location of data points.

In particular, given a target point $p_0 \in \mathbb{R}^2$ and a set of $nk$ points $P_{\text{flat}} = \uplus_{i=1}^n P_i$, where $P_i = \{p_{i,1}, p_{i,2}, \cdots, p_{i,k}\} \subset \mathbb{R}^2$, we use simulation to show that we only need to change $\tilde{n}\tilde{k}$ points of $P_{\text{flat}}$, then we can get a new set $\widetilde{P}_{\text{flat}} = \uplus_{i=1}^n \widetilde{P}_i$ such that $\text{median}(\text{median}(\widetilde{P}_1), \text{median}(\widetilde{P}_2), \cdots, \text{median}(\widetilde{P}_n)) = p_0$. Here, the "median" means $L_1$-median, and

$$\tilde{n} = \begin{cases} \frac{1}{2}n & \text{if } n \text{ is even} \\ \frac{1}{2}(n+1) & \text{if } n \text{ is odd} \end{cases}, \quad \tilde{k} = \begin{cases} \frac{1}{2}k & \text{if } k \text{ is even} \\ \frac{1}{2}(k+1) & \text{if } k \text{ is odd} \end{cases}.$$

To do this, we first show that, given $k$ points $S = \{(x_i, y_i) \mid 1 \leq i \leq k\}$ in $\mathbb{R}^2$, and a target point $(x_0, y_0)$, we can change $\tilde{k}$ points of $S$ to make $(x_0, y_0)$ as the $L_1$-median of the new set. As $n$ and $k$ grow, then $\tilde{n}\tilde{k}/(nk) = 0.25$ is the asymptotic breakdown point of this estimator, as a consequence of Theorem 2, and thus we may need to move this many points to get the result.

If $(x_0, y_0)$ is the $L_1$-median of the set $\{(x_i, y_i) \mid 1 \leq i \leq k\}$, then we have [13]:

$$\sum_{i=1}^k \frac{x_i - x_0}{\sqrt{(x_i - x_0)^2 + (y_i - y_0)^2}} = 0, \quad \sum_{i=1}^k \frac{y_i - y_0}{\sqrt{(x_i - x_0)^2 + (y_i - y_0)^2}} = 0. \tag{13}$$

We define $\vec{x} = (x_1, x_2, \cdots, x_{\tilde{k}})$, $\vec{y} = (y_1, y_2, \cdots, y_{\tilde{k}})$ and

$$h(\vec{x}, \vec{y}) = \left(\sum_{i=1}^k \frac{x_i - x_0}{\sqrt{(x_i - x_0)^2 + (y_i - y_0)^2}}\right)^2 + \left(\sum_{i=1}^k \frac{y_i - y_0}{\sqrt{(x_i - x_0)^2 + (y_i - y_0)^2}}\right)^2.$$

Since (13) is the sufficient and necessary condition for $L_1$-median, if we can find $\vec{x}$ and $\vec{y}$ such that $h(\vec{x}, \vec{y}) = 0$, then $(x_0, y_0)$ is the $L_1$-median of the new set.

Since

$$\partial_{x_i} h(\vec{x}, \vec{y}) = 2\left(\sum_{j=1}^k \frac{x_j - x_0}{\sqrt{(x_j - x_0)^2 + (y_j - y_0)^2}}\right) \frac{(y_i - y_0)^2}{\left((x_i - x_0)^2 + (y_i - y_0)^2\right)^{\frac{3}{2}}}$$

$$- 2\left(\sum_{j=1}^k \frac{y_j - y_0}{\sqrt{(x_j - x_0)^2 + (y_j - y_0)^2}}\right) \frac{(x_i - x_0)(y_i - y_0)}{\left((x_i - x_0)^2 + (y_i - y_0)^2\right)^{\frac{3}{2}}},$$

$$\partial_{y_i} h(\vec{x}, \vec{y}) = -2\Big(\sum_{j=1}^{k} \frac{x_j - x_0}{\sqrt{(x_j - x_0)^2 + (y_j - y_0)^2}}\Big) \frac{(x_i - x_0)(y_i - y_0)}{\big((x_i - x_0)^2 + (y_i - y_0)^2\big)^{\frac{3}{2}}}$$

$$+ 2\Big(\sum_{j=1}^{k} \frac{y_j - y_0}{\sqrt{(x_j - x_0)^2 + (y_j - y_0)^2}}\Big) \frac{(x_i - x_0)^2}{\big((x_i - x_0)^2 + (y_i - y_0)^2\big)^{\frac{3}{2}}},$$

we can use gradient descent to compute $\vec{x}, \vec{y}$ to minimize $h$. For the input $S = \{(x_i, y_i) | 1 \le i \le k\}$, we choose the initial value $\vec{x}_0 = \{x_1, x_2, \cdots, x_{\tilde{k}}\}$, $\vec{y}_0 = \{y_1, y_2, \cdots, y_{\tilde{k}}\}$, and then update $\vec{x}$ and $\vec{y}$ along the negative gradient direction of $h$, until the Euclidean norm of gradient is less than 0.00001.

The algorithm framework is then as follows, using the above gradient descent formulation at each step. We first compute the $L_1$-median $m_i$ for each $P_i$, and then change $\tilde{n}$ points in $\{m_1, m_2, \cdots, m_n\}$ to obtain $\{m_1', m_2', \cdots, m_{\tilde{n}}', m_{\tilde{n}+1}, \cdots, m_n\}$ such that $\mathrm{median}(m_1', m_2', \cdots, m_{\tilde{n}}', m_{\tilde{n}+1}, \cdots, m_n) = p_0$. For each $m_i'$, we change $\tilde{k}$ points in $P_i$ to obtain $\widetilde{P}_i = \{p_{i,1}', p_{i,2}', \cdots, p_{i,\tilde{k}}', p_{i,\tilde{k}+1}, \cdots, p_{i,k}\}$ such that $\mathrm{median}(\widetilde{P}_i) = m_i'$. Thus, we have

$$\mathrm{median}\big(\mathrm{median}(\widetilde{P}_1), \cdots, \mathrm{median}(\widetilde{P}_{\tilde{n}}), \mathrm{median}(P_{\tilde{n}+1}), \cdots, \mathrm{median}(P_n)\big) = p_0. \qquad (14)$$

To show a simulation of this process, we use a uniform distribution to randomly generate $nk$ points in the region $[-10, 10] \times [-10, 10]$, and generate a target point $p_0 = (x_0, y_0)$ in the region $[-20, 20] \times [-20, 20]$, and then use our algorithm to change $\tilde{n}\tilde{k}$ points in the given set, to make the new set satisfy (14). Table 1 shows the result of running this experiment for different $n$ and $k$, where $(x_0', y_0')$ is the median of medians for the new set obtained by our algorithm. It lists the various values $n$ and $k$, the corresponding values $\tilde{n}$ and $\tilde{k}$ of points modified, and the target point and result of our algorithm. If we reduce the terminating condition, which means increasing the number of iteration, we can obtain a more accurate result, but only requiring the Euclidean norm of gradient to be less than 0.00001, we get very accurate results, within about 0.01 in each coordinate.

We illustrate the results of this process graphically for a example in Table 1: for the cases $n = 5$,

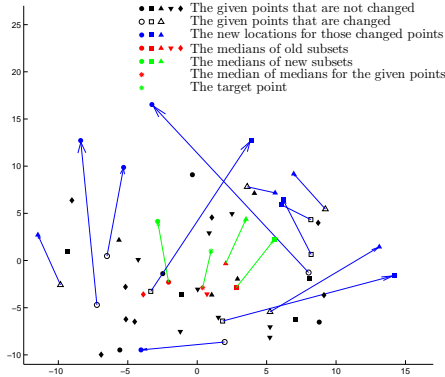

● ■ ▲ ▼ ◆ The given points that are not changed
○ □ △ The given points that are changed
● ■ ▲ The new locations for those changed points
■ ▲ ▼ ◆ The medians of old subsets
● ■ ▲ The medians of new subsets
● The median of medians for the given points
● The target point

Figure 1: The running result for the case $n = 5$, $k = 8$, $(x_0, y_0) = (0.99, 1.01)$ in Table 1.

| $n$ | $k$ | $\tilde{n}$ | $\tilde{k}$ | $(x_0, y_0)$ | $(x_0', y_0')$ |
|---|---|---|---|---|---|
| 5 | 8 | 3 | 4 | (0.99, 1.01) | (0.99, 1.01) |
| 5 | 8 | 3 | 4 | (10.76, 11.06) | (10.70 11.06) |
| 10 | 5 | 5 | 3 | (-13.82, -4.74) | (-13.83, -4.74) |
| 50 | 20 | 25 | 10 | ( -14.71, -13.67) | (-14.72, -13.67) |
| 100 | 50 | 50 | 25 | ( -14.07, 18.36) | ( -14.07, 18.36) |
| 500 | 100 | 250 | 50 | (-15.84, -6.42) | (-15.83, -6.42) |
| 1000 | 200 | 500 | 100 | (18.63, -12.10) | (18.78, -12.20) |

Table 1: The running result of simulation.

$k = 8$, $(x_0, y_0) = (0.99, 1.01)$, wihch is shown in Figure 1. In this figure, the green star is the target point. Since $n = 5$, we use five different markers (circle, square, upward-pointing triangle, downward-pointing triangle, and diamond) to represent five kinds of points. The given data $P_{\mathsf{flat}}$ are shown by black points and unfilled points. Our algorithm changes those unfilled points to the blue ones, and the green points are the medians of the new subsets. The red star is the median of medians for $P_{\mathsf{flat}}$, and other red points are the median of old subsets. So, we only changed 12 points out of 40, and the median of medians for the new data set is very close to the target point.

## 5  Conclusion

We define the breakdown point of the composition of two or more estimators. These definitions are technical but necessary to understand the robustness of composite estimators. Generally, the composition of two of more estimators is less robust than each individual estimator. We highlight a few applications and believe many more exist. These results already provide important insights for complex data analysis pipelines common to large-scale automated data analysis.

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
