[Reviews · NeurIPS 2016]

Reviewer 1

Summary

The authors study the effect of composition on the robustness of estimators. To do this they use/define the notion of a breakdown point. The main theorem state that the breakdown point of a composite estimator E (= E2 o E1) is larger (and in some cases equal) to the product of the breakdown point of E2 and E1.

Qualitative Assessment

1) The text is a bit difficult to follow. I think that the main conclusion is that the composition of two robust estimators is not robust. This is stated mostly at the very end of the text (in the conclusion). I think it should appear in the introduction. Example 2 and 4 in the application illustrate this point, I think they should be before example 1 and 3. 2) In line 183 the authors say "we need the definition of asymptotic onto-breakdown point" From Theorem 2 it is not clear to me that this a necessary condition ? 3) To choose an estimator it is important to look at its robustness. Other criteria are also important (MSE, bias...) and sometimes robustness has a cost in terms of MSE, power... This is not discussed by the authors. I think it should. In particular, I think example 1 and 3 of the application section would be clearer and more convincing if the authors provided the MSE, bias... of the various estimators they study. 4) A number of statements in the text are blunt and written without proofs or references. For example line 25-27: "...these unwanted outcomes cannot occur" line 44: "...they are not mathematically rigourous" line 296: "...that can likely be accounted in other ways" 5) I think a number of statements in the text are not appropriate for a scientific paper line 242 : "not the executives who may have exorbitant pay" 6) Proofs should be in appendix or at the end of the paper.

Confidence in this Review

2-Confident (read it all; understood it all reasonably well)


Reviewer 2

Summary

In their contribution, the authors study the robustness of composite estimators. In particular they find that classical definitions of breakdown are inappropriate for composite estimators, and introduce the "Asymptotic Onto-Breakdown Point", which is shown to be a more appropriate notion of robustness.

Qualitative Assessment

In this very well written presentation, the authors study the (asymptotic) breakdown of composite estimators. The problem is an interesting one, and the analysis very elegant. I find, however, that this result belongs in the robust statistics literature, more than it belongs in NIPS. Perhaps if the authors had provided more convincing applications, this would not have been the case.

Confidence in this Review

2-Confident (read it all; understood it all reasonably well)


Reviewer 3

Summary

This paper is concerned with the robustness of composed estimators. After defining the breakdown point of an estimator that measures its robustness to outliers, theorems show the relation between the breakdown point of the composed estimator and the breakdown points of the individual estimators. Then, a lot of examples are provided.

Qualitative Assessment

I like this paper, which is concerned with an important and interesting problem, the results are nice. I did not give better marks because most of the paper only illustrates the first two theorems (by the way, I think Section 2.3 is straightforward, and at least the proof of Theorem 3 should be omitted): I think the quantity of results is not totally satisfying.

Confidence in this Review

2-Confident (read it all; understood it all reasonably well)


Reviewer 4

Summary

The paper considers the robustness of composite estimators, estimators that apply one estimator to sections of the data set and another estimator to the output of the first estimator. The problem is important due to the fact that big data sets are hard to curate and composite estimators are often applied to them to simplify data analysis. The notion of robustness is quantified by the breakdown point, which is the minimum fraction of data points that need to be significantly modified in order to significantly modify the output. The breakdown point of a composite estimator is usually the product of the breakdown points of its component estimators. The authors apply this result to analyze the robustness of existing composite estimators (such as median of medians) and propose ways to increase their robustness.

Qualitative Assessment

Technical Quality: The technical quality is good. The proofs appear to be sound and the remarks touch on important issues. The main text does not give experimental results. My main concern is how to couple this work with qualities of estimators that are traditionally desirable in statistics. For example, people often want unbiasedness or statistical efficiency. The applications section talks about how to make estimators more robust, but can we get estimators that are robust and have a low MSE? Novelty: The paper is fairly novel. It builds upon previous basic work on the robustness of estimators via the breakdown point by defining a new version of the breakdown point and extending to composite estimators. Potential impact: I think that this work is important for real-world data analysis. In some fields existing algorithms have many steps giving complicated composite estimators, but the problem of error propagation and robustness may not be given due attention. This work may also be useful for companies that don't spend the time to curate their business data. My only concern is the difficulty of computing the breakdown point for more complicated estimators. The paper only considers percentiles and other median-based estimators. Clarity and presentation: The paper is well-organized and well-written, and the authors communicate the ideas clearly. The proofs were easy to follow. However, there are a few typos and grammar errors. Also, to better show the applicability of the work, I think that the simulations ought to be added to the paper and the complete proofs pushed to the supplement if needed.

Confidence in this Review

2-Confident (read it all; understood it all reasonably well)


Reviewer 5

Summary

This paper studies the robustness of composite estimators, where in the first step, an estimator is applied to subsets of the data, and then a second estimator is applied the output of the first step. Robustness is defined using a technical "finite sample breakdown point", essentially representing the number of unbounded data points before the estimator output becomes unbounded. Conditions are given for when the breakdown point of the composite estimator equals the product of the individual breakdown points. The results are extended to more than two-level estimators, and they conclude with some applications.

Qualitative Assessment

The paper is a readable introduction to the "breakdown point" theory of estimator robustness. It is overall well-written and provides some intuition for what these breakdown points mean. I am uncertain as to the overall level of interest in the work/impact for the NIPS audience, however. The theorems are nicely motivated with examples, and the breakdown points of a few estimators (L1-median, Siegel) are given. However, it would be nice to see the authors work through and example calculation of how to compute the breakdown point of those estimators. It would also be nice to see a numerical example, e.g. for one of the applications given towards the end. Then, the reader could verify that the breakdown point is applicable for real or realistic data. A few other comments: p.1, typo: "roll" should be "role" pg.2, It would be nice to mention which of these non-rigorous breakdown definitions is closest to your eventual definition. You use the set P_flat throughout, but I'm not sure what "flat" refers to. I'd explain more or change the notation. What does the union symbol with the plus sign mean?

Confidence in this Review

2-Confident (read it all; understood it all reasonably well)


Reviewer 6

Summary

The paper introduced the concept of asymptotic Onto-Breakdown Point, and showed the breakdown point of the composite estimator is the product of the breakdown points of the individual estimators.

Qualitative Assessment

1. Whether the restriction that sample size of each subdata $P_i$ has to be the same can be relaxed? 2. Equation 17 asks for the equality of asymptotic breakdown point and asymptotic onto- breakdown point, is there a way to check this condition for a specific estimator? 3. Breakdown point is most useful in small sample situations, since it's simplicity and straightforwardness. It is important to treat the breakdown point as a finite sample concept, equation (10) gave a lower bound of the finite sample breakdown point. For theorem 2, if the results can be shown for all sample size, it would be more interesting. 4. In application 1, for E_1, as shown in Remark 2, the asymptotic breakdown point (0.45) and asymptotic onto-breakdown point (0.55) doesn't equal, which violates the assumption in equation (17) for theorem 2, please clarify how still the breakdown point of E equals to 0.45*0.5 = 0.225.

Confidence in this Review

3-Expert (read the paper in detail, know the area, quite certain of my opinion)